# Near-ideal spontaneous photon sources in silicon quantum photonics

S. Paesani[1,4], M. Borghi[1,2,4], S. Signorini [3,4], A. Maïnos [1], L. Pavesi[3] & A. Laing [1✉]

While integrated photonics is a robust platform for quantum information processing, architectures for photonic quantum computing place stringent demands on high quality information carriers. Sources of single photons that are highly indistinguishable and pure, that are either near-deterministic or heralded with high efficiency, and that are suitable for mass-manufacture, have been elusive. Here, we demonstrate on-chip photon sources that simultaneously meet each of these requirements. Our photon sources are fabricated in silicon using mature processes, and exploit a dual-mode pump-delayed excitation scheme to engineer the emission of spectrally pure photon pairs through inter-modal spontaneous four-wave mixing in low-loss spiralled multi-mode waveguides. We simultaneously measure a spectral purity of $0.9904 \pm 0.0006$, a mutual indistinguishability of $0.987 \pm 0.002$, and >90% intrinsic heralding efficiency. We measure on-chip quantum interference with a visibility of $0.96 \pm 0.02$ between heralded photons from different sources.

[1] Quantum Engineering Technology Labs, H. H. Wills Physics Laboratory and Department of Electrical and Electronic Engineering, University of Bristol, Bristol BS81FD, UK. [2] SM Optics s.r.l., Research Programs, Via John Fitzgerald Kennedy 2, 20871 Vimercate, Italy. [3] Department of Physics, University of Trento, Via Sommarive 14, 38123 Trento, Italy. [4] These authors contributed equally: S. Paesani, M. Borghi, S. Signorini. ✉email: anthony.laing@bristol.ac.uk

Sustained progress in the engineering of platforms for quantum information processing has recently achieved a scale that surpasses the capabilities of classical computers to solve specialised and abstract problems[1–4]. But while achieving a computational advantage for practical or industrially relevant problems may be possible with further scaling of special purpose NISQ (noisy intermediate scale quantum) devices[1], more general purpose quantum computers will require a hardware platform that integrates millions of components, individually operating above some fidelity threshold[5,6]. Silicon quantum photonics[7], which is compatible with complementary metal-oxide-semiconductor (CMOS) fabrication, provides a potential platform for very large-scale quantum information processing[8–10].

All-photonic quantum computing architectures rely on arrays of many photon sources to achieve combinatorial speed-ups in quantum sampling algorithms[11,12], or to approximate an on-demand source of single photons[13–15], and supply entangling circuitry for general purpose quantum computing[8,16]. In the former case, the level of indistinguishability among photons upper bounds the computational complexity of sampling algorithms[17]; in the latter case, photon impurity and distinguishability lead to logical errors[16,18]. Furthermore, and in general, lossy or inefficient heralding of photons vitiates the scaling of photonic quantum information processing. The lack of a demonstration that simultaneously overcomes all of these challenges has been a bottleneck to scalability for quantum computing in integrated photonics.

Progress in solid-state sources of single photons make quantum dots an attractive approach for certain NISQ experiments[19,20]. However, the low-loss integration of solid-state sources into photonic circuitry, that maintains distinguishability over many photons, is an ongoing challenge[21,22]. Integrated sources of photons based on spontaneous processes, such as four-wave mixing (SFWM) in single-mode waveguides or micro-ring cavities[7,23] are appealing for their manufacturability. However, spontaneous sources incur limitations to purity and to heralding efficiency[23], with micro-ring cavities additionally requiring resonance tuning to avoid distinguishability among different cavities[24,25].

Here, we demonstrate the engineering of a CMOS-compatible source of heralded single photons using silicon photonics, which simultaneously meets the requirements for scalable quantum photonics: high purity, high heralding efficiency, and high indistinguishabilty.

The source is based on inter-modal SFWM, where phase-matching is engineered by propagating the pump in different transverse modes of a spiralled multi-mode (MM) waveguide[26].

## Results

### Discrete-band inter-modal SFWM phase-matching and delayed-pump scheme.
Integrated photon sources in silicon are based on SFWM, where, if phase-matching (momentum conservation) and energy conservation conditions are satisfied, light from a pump laser can be converted into pairs of single photons[23,27]. In standard SFWM in single-mode waveguides, near-zero dispersion produces broad phase-matching bands around the pump wavelength, such that the process, dominated by energy conservation conditions, induces undesired strong spectral anticorrelations between the emitted photons. In contrast, in this work we suppress such correlations adopting a inter-modal approach to SFWM. As shown in Fig. 1a, an input pulsed laser coherently pumps the two lowest order transverse magnetic (TM) modes of a MM waveguide, namely TM0 and TM1 (see Fig. 1a inset), and generates pairs of idler and signal photons in these modes via inter-modal phase-matching. The dispersion

relations between the TM0 and TM1 modes are such that a discrete narrow phase-matching band appears[26]. By tailoring the waveguide cross-section, the modal dispersion can be accurately engineered to design the phase-matching band with a bandwidth similar to the pump bandwidth (related to energy conservation). This suppresses the frequency anticorrelations imposed by energy conservation, and enhances the spectral purity of the emitted photons[26]. In particular, we exploit the different modal group velocities in silicon waveguides to achieve a condition where the idler and signal photons are generated on TM0 at $\simeq 1516$ nm and on TM1 at $\simeq 1588$ nm, respectively, with a bandwidth of approximately 4 nm (see Supplementary Note 1).

Moreover, to obtain a near-unit spectral purity, we further suppress residual correlations in the joint spectrum by inserting a delay $\tau$ on the TM0 component of the pump (with higher group velocity than TM1) before injecting it in the source. The delay gradually increases and decreases the temporal overlap between the pump in the TM0 and TM1 modes along the multi-modal waveguide source (colour-coded in Fig. 1a). This results in an adiabatic switching of non-linear interactions in the source, which suppresses spurious spectral correlations[28,29]. Simulations (see Supplementary Note 1) predict a spectral purity of 99.4% in this configuration, in contrast to the case where no delay is applied, which predicts a purity of 84.0%, as shown in Fig. 1b.

**Source design.** Figure 1c, shows the compact footprint for the MM waveguide source obtained by adopting a spiral geometry. The delayed-pump excitation scheme is implemented in three stages, as shown in Fig. 1a. The pump, initially in TM0, is split by a balanced beam-splitter; one arm receives a delay of $\tau$ with respect to the other, then the two arms are recombined using a TM0 to TM1 mode converter, and injected in the MM waveguide. Once generated, the signal photon is separated from the idler via a second TM1 to TM0 mode converter. After processing, signal and idler photons are out-coupled to fibres, where the pump is filtered out via broad-band fibre bragg-gratings, and single photons are detected with superconducting-nanowire single photon detectors (SNSPDs).

**Source brightness characterisation.** We experimentally characterised the squeezing value $\xi$ of the generated two-mode squeezed state emitted from individual sources with second-order correlation measurements[30] (see Supplementary Note 5). Fig. 1e compares results for both the delayed and the non-delayed cases. Experimental results confirm our simulations and additionally demonstrate higher brightness as a benefit of the temporal delay scheme (see Supplementary Note 1). Squeezing values up to $|\tanh(\xi)|^2 \simeq 0.2$ are observed using a small input (off-chip) average pump power of 3 mW, corresponding to >8 MHz photon-pair generation rates on-chip. To reduce noise from multi-photon events, measurements reported from this point on are performed with an input pump power of approximately 0.5 mW: coincidence rates are measured at 15 kHz, with heralded single photon $g_h^{(2)}$ measured at 0.053(1) (Fig. 1e inset).

**Spectral purity characterisation.** Source purity is first estimated from a direct measurement of the joint spectral intensity (JSI)[31]. The JSI reconstruction is implemented using narrow-bandwidth tunable filters to scan the emitted wavelengths of the signal and idler photons, as pictured in Fig. 1f. Data from a source with no temporal delay yields a JSI with a spectral purity of 93.1(2)%, which increases to 99.04(6)%, in the scheme with a delay, as shown in Fig. 1g. The contrasting measurements show a clear suppression of spurious correlations with the delay scheme. A second estimation of the emitted single photon purity

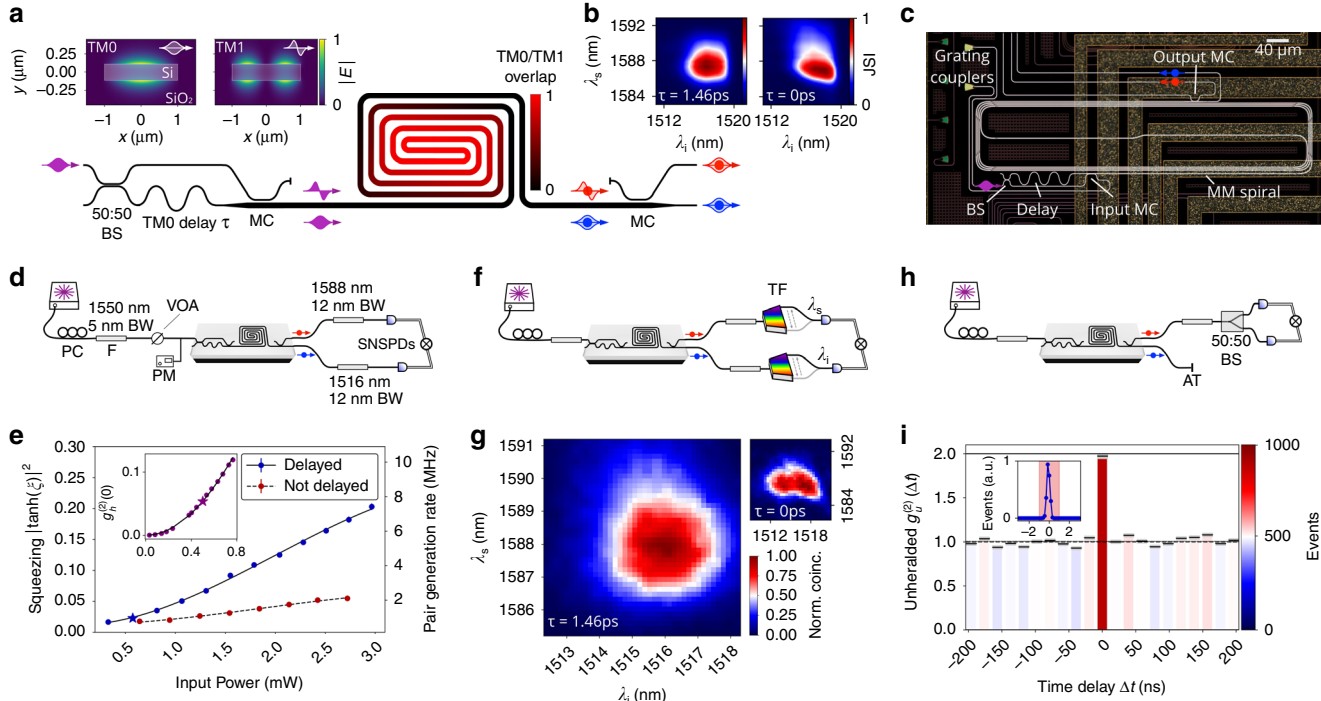

**Fig. 1 Design and performance of the multi-modal source. a** Schematic of the source. An input near-1550 nm pulsed pump laser (4.5 nm bandwidth), initially propagating in the TM0 mode, is split using a 50:50 beam-splitter (BS). The output in the upper arm of the BS is converted into the TM1 mode via a mode-converter (MC), while the TM0 output in the lower arm is delayed by a time $\tau = 1.46$ ps. Due to the different group velocities, the two modes become overlapped and subsequently diverge again while propagating through the source, as qualitatively colour-coded in the figure. Photon pairs, with the signal photon (near 1588 nm) in the TM1 mode and the idler photon (near 1516 nm) in the TM0 mode, are emitted via inter-modal SFWM and finally deterministically separated via a MC. Inset: cross sections of the TM0 and TM1 modes in the MM waveguide. **b** Simulated JSI of the source in the presence of a delay $\tau = 1.46$ ps (left) and with no delay (right), with corresponding single photon purities of 99.4% and 84.0%. **c** Optical microscope image of a single multi-modal source structure-waveguides are highlighted. **d** Set-up to characterise squeezed light via second-order correlation measurements, using a polarisation controller (PC), fibre pass-band filter (F), variable optical attenuator (VOA), and an optical power monitor (PM). **e** Measured squeezing as a function of (off-chip) pump power. Blue and red points are data measured in a source with and without delay, respectively, with a fit shown as a black line. The stars indicate the typical operating regime. Inset: measured heralded $g_h^{(2)}(0)$ as a function of input powers. **f** Set-up for the characterisation of the emitted JSI, using a tunable filter (TF). **g** Measured JSI from the source with delay (left) and without delay (right), with respective corresponding spectral purities of 0.9904(6) and 0.931(2). **h** Set-up for purity characterisation via unheralded second-order correlation measurements. Idler photons are discarded via an absorbing termination (AT). **i** Measured unheralded $g_u^{(2)}(\Delta t)$ in the source with delay. Each bar corresponds to a coincidence window of 2 ns (inset). The measured $g_u^{(2)}(0) = 1.97(3)$ corresponds to a photon spectral purity of 0.97(3). Error bars represent 1 s.d. and are calculated assuming Poissonian error statistics.

for the delayed structure is performed via unheralded second-order correlation measurements $g_u^{(2)}$ [30]. These are implemented by dividing the output signal mode with an off-chip balanced fibre beam-splitter and measuring coincidences between the two output arms (see Fig. 1h). Measured unheralded second order-correlation values are reported in Fig. 1i. We obtain $g_u^{(2)}(0) = 1.97(3)$, which corresponds to a single photon purity of 97(3)%, consistent with the value obtained from the JSI.

**Heralding efficiency characterisation**. The capability of the sources to generate pure photons with no requirement for filtering enables the simultaneous achievement of high heralding efficiency and high purity. In our experiment, off-chip filters are used solely for pump rejection: their bandwidth (12 nm, flat transmission) contains >99% of the emitted spectra, which results in ultra-high filtering heralding efficiency [32]. While the effect of filtering is thus negligible, the intrinsic heralding efficiency of the source is affected by linear and non-linear transmission losses inside the waveguide. These losses are, however, greatly mitigated in MM waveguides (which present <0.5 dB/cm linear loss, see Supplementary Note 2). Taking into account the characterised losses, we estimate a heralding efficiency of approximately 95%

for an individual source. The measured heralding efficiency at the off-chip detectors is 12.6(2)%, corresponding to 91(9)% on-chip intrinsic heralding efficiency after correcting for the characterised losses in the channel to the detectors (see Supplementary Note 2), which can be highly suppressed by implementing low-loss off-chip couplers [33] or with integrated detectors [34].

**Source indistinguishability characterisation**. To experimentally test the source indistinguishabilty we integrate a reconfigurable photonic circuit to perform quantum interference between different sources. Schematics of the circuit are shown in Fig. 2a–b. Two sources are coherently pumped by splitting the input laser with an on-chip tunable Mach-Zehnder interferometer (MZI); the resulting idler and signal modes from the different sources are grouped and interfered on-chip using additional integrated phase-shifters and MZIs (see Methods). Using this circuit, we experimentally estimate the indistinguishability among the sources using three different types of measurements. First, we reconstruct the JSI of each source by operating the two sources individually. The overlap of the JSIs reconstructed from each source (Fig. 2c) estimates a mutual indistinguishability of 98.5(1)%.

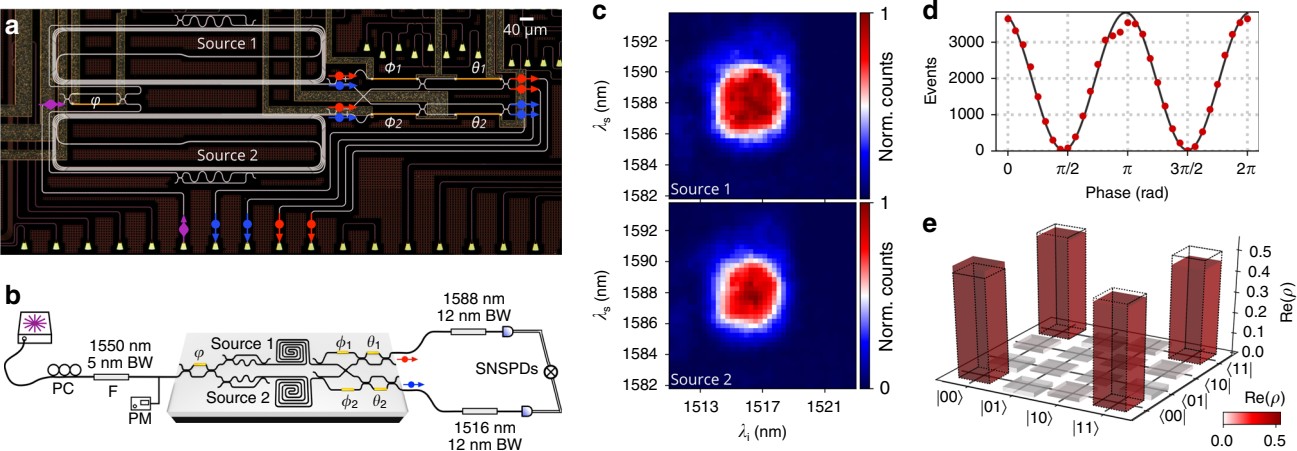

**Fig. 2 Multiple sources and indistinguishability characterisation. a** Optical microscope image of the device for the coherent pumping of two sources and processing of the emitted photons. Input pump light is split between the two sources via an input MZI. Waveguide crossings are used to group the signal and idler photons emitted from both sources, and arbitrary unitary operations on signals (idlers) are implemented via a phase-shifter $\phi_1$ ($\phi_2$) and a MZI with internal phase $\theta_1$ ($\theta_2$). **b** Schematic of the integrated circuit and set-up to characterise the indistinguishability between the sources. **c** Individual JSIs measured with separate pumping of source 1 (top panel) and source 2 (bottom panel). The indistinguishability of the two measured spectra, calculated with the overlap of the two JSIs, is 0.985(1). **d** Measured reverse-HOM fringe from the two sources. Error bars, from Poissonian photon statistics, are smaller than markers. The fringe visibility, which corresponds to source indistinguishability, is 0.987(2). **e** Density matrix of the two-qubit entangled state obtained when coherently pumping the two sources, reconstructed via quantum state tomography. Fidelity with the ideal state $|\Phi_+\rangle = (|00\rangle + |11\rangle)/\sqrt{2}$, pictured as transparent bars, is 0.989(3). The source indistinguishability inferred from the tomography is 0.982(6).

A second measurement of the indistinguishability was performed via reversed Hong-Ou-Mandel (HOM) interference between the two sources[9,35,36]. Both sources were pumped and the respective idler and signal modes were interfered by tuning the output MZIs to act as 50:50 beam-splitters. The 98.7(2)% visibility of the reversed HOM fringe, shown in Fig. 2d and obtained by scanning the phases $\phi_1 = \phi_2 = \phi$, corresponds directly to the source indistinguishability (see Supplementary Note 6).

A further estimate of indistinguishability is obtained by testing the entanglement generated when coherently pumping the two sources[9,36]. Using quantum state tomography, we experimentally reconstruct the density matrix shown in Fig. 2e, which has a fidelity of 98.9(3)% with the ideal maximally-entangled state $|\Phi_+\rangle = (|00\rangle + |11\rangle)/\sqrt{2}$, and provides an indistinguishability value of 98.2(6)% (see Supplementary Note 6 for details).

**Heralded Hong-Ou-Mandel experiments.** A key figure of merit for multi-photon experiments, particularly in the context of many photon quantum information processing, is the heralded Hong-Ou-Mandel visibility, which quantifies the interference of photons heralded from different sources. This quantity, which simultaneously incorporates source indistinguishability, purity and absence of multi-photon noise, determines the stochastic noise in photonic quantum computing architectures[8,18], and the computational complexity achievable in photonic sampling algorithms[17]. We implemented heralded HOM experiments by operating our two-source device in the four-photon regime, as shown in Fig. 3a. The circuit is configured such that idler photons from both sources are directly out-coupled to detectors to herald the signal photons, which are interfered in the MZI (see inset of Fig. 3b). The heralded HOM fringe is measured by scanning the phase $\theta_1$ inside the MZI and collecting 4-photon events[37,38]. The measured on-chip heralded HOM fringe is shown in Fig. 3b. The raw-data visibility (no multi-photon noise correction) is 96(2)%.

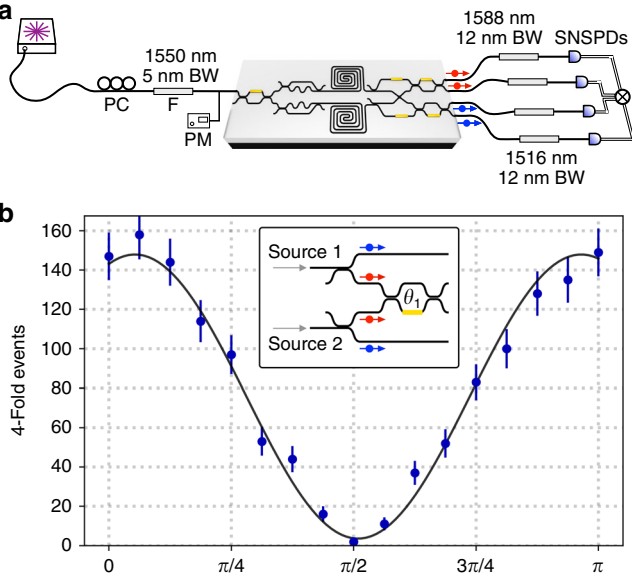

**Fig. 3 Heralded Hong-Ou-Mandel interference. a** Experimental setup for 4-photon heralded HOM experiments. **b** Heralded HOM results. Points are raw four-photon counts measured over 4-h of integration time each for different values of the MZI phase $\theta$, with a solid line fit to the data, presenting a visibility of 0.96(2). Error bars represent 1 s.d. and consider Poissonian photon statistics. Inset: schematic of the integrated circuit configuration for measuring the heralded HOM fringe.

## Discussion
Our results have a significant impact on the prospects of quantum information processing in integrated photonics. Photon sources from previous state-of-the-art integrated photonic devices demonstrated an on-chip heralded quantum inference raw

visibility of 82%[38] which upper-bounds any potential quantum sampling experiment to a computational complexity equivalent to 31-photon interference (considering error bounds of 10%[17]). Our results lift this bound to a computational complexity equivalent to >150 photon interference (see Supplementary Note 4), deep in the regime of quantum computational supremacy[39].

Furthermore, in the context of digital quantum computing, our results make a significant leap toward the ≳99.9% heralded HOM visibility required to construct lattices of physical qubits with error rates below 1% using current fault-tolerance photonic architectures[16,18]. Our analysis (see Supplementary Note 3) suggests that heralded HOM visibilities of 99.9% could be achievable with minor modifications to our design; for example by using an improved quality pump laser and by using semiconductor fabrication processes with approximately 4 nm uniformity[40,41]. Our results represent the near removal of a critical set of physical errors that had limited the scaling of photonic quantum information processing.

## Methods

**Device fabrication.** The silicon devices used were fabricated using CMOS-compatible UV-lithography processes in a commercial multi-project wafer run by the Advanced Micro Foundry (AMF) in Singapore. Waveguides are etched in a 220 nm silicon layer atop a 2 μm buried oxide, and an oxide top cladding of 3 μm. The thermo-optic phase-shifters to reconfigure the integrated circuits are formed by TiN heaters positioned 2 μm above the waveguide layer.

**Inter-modal four-wave mixing in silicon waveguides.** Inter-modal spontaneous four-wave mixing is performed by propagating the pump, signal and idler waves on different waveguide modes. The spectral properties of the perfect phase matching depend on the group velocity dispersion of the different modes employed in the process, which can be tuned by engineering the waveguide geometry. In our experiment, we operate the pump on the TM0 and TM1 modes, and the signal and idler on the TM1 and TM0, respectively. With these modes, phase matching of the SFWM process is enabled in the 1500–1600 nm spectral window using standard silicon-on-insulator waveguides with a geometry of 2 μm × 0.22 μm, which is used in our source design.

**Pump-delayed generation.** When a delay is applied between non-degenerate pump pulses, the effective interaction length depends on the value of such delay. This is due to the walk-off, which limits the nonlinear length. The best scenario is when the overtaking process of the faster pulse over the slower one occurs completely within the waveguide, thus maximising the interaction length and the generation efficiency. In this case the delay is such that the maximum spatial overlap between the pump pulses occurs in the middle of the waveguide. With different delays, the nonlinear medium is not optimally exploited, resulting in reduced generation efficiency. The delay used to optimise the generation efficiency corresponds to the delay for maximum spectral purity (details in Supplementary Note 1).

**Source design.** The 2 μm × 0.22 μm multi-mode waveguide in the source is designed with a length of 11 mm and an initial temporal delay of $\tau = 1.46$ ps between the TM0 and TM1 modes. A spiral geometry for the waveguide is used to increase the compactness. Modal cross-talk in the spiral is kept below $-25$ dB extinction by adopting 90° Euler bends of radius 45 μm (see Supplementary Note 1 for more details). The footprint of an individual silicon-on-insulator source with our design is approximately 200 μm × 900 μm. The TM0–TM1 mode converters used to inject the pump in the MM waveguide and separate the signal and idler photons at the output have $< -30$ dB characterised modal cross-talk, and >95% conversion efficiency.

**Integrated circuit.** The integrated circuit pictured in Fig. 2a (see also Supplementary Fig. 6 for a more detailed schematic) used for the multi-source interference experiments consists of three reconfigurable MZIs (internal phases $\varphi$, $\theta_1$ and $\theta_2$), two phase-shifters ($\phi_1$, $\phi_2$), a broad-band waveguide crosser, and two sources. The circuit used is a two-mode version of the circuits implemented, for example, in ref. [9]. At the input, the MZI $\varphi$ is configured to split the pump between the two sources: using $\varphi = 0$ ($\varphi = \pi$) we operate the sources individually by pumping only source 1 (source 2), while $\varphi = \pi/2$ implements a balanced pump splitting to coherently operate both sources simultaneously. When uniformly pumping both sources ($\varphi = \pi/2$), each source receives half of the input pump power, and thus the photon-pair generation probability in each source is decreased by a factor four compared to the single source regime. After photons are generated in the sources, the waveguide crosser allows us to route together to signal and idler

modes. Arbitrary and reconfigurable two-mode unitary operations are then performed on the signal (idler) modes via the phase $\phi_1$ ($\phi_2$) and the MZI $\theta_1$ ($\theta_2$). Light is coupled in and out of the circuit by means of TM0 focusing grating couplers, which have been individually optimised to maximise their efficiency at the pump, signal and idler wavelengths ($\simeq 6.6$ dB loss per coupler). The coupling was observed to be stable over few hours, but to gradually decrease over longer periods (approximately between 0.5 and 1 dB/day without active coupling optimisation). Insertion losses for the individual on-chip circuit components in our devices are: 0.19 dB/cm (0.40 dB/cm) for TM0 (TM1) transmission in the MM waveguide, 0.1 dB per mode converter, <0.01 dB loss per directional coupler, and 0.4 dB per waveguide crossing.

Total insertion losses in the integrated circuit are approximately 14 dB, mostly due to grating couplers. In both the single-source circuit (Fig. 1b) and the two-source circuit (Fig. 2a) the intrinsic heralding efficiencies of the sources have been measured to be approximately the same, with a near-90% efficiency after correcting for the off-chip channel loss. Off-chip pump rejection filters have an insertion loss for the unfiltered photons of 0.4 dB. When including this external transmission loss for pump rejection, the heralding efficiency of the systems is approximately 83%. See Supplementary Note 2 for more details on the design and characterisation of the individual components.

**Experimental set-up.** Pump pulses at 1550 nm (4.5 nm bandwidth, 800 fs pulse length, 50 MHz repetition rate) from an erbium-doped fibre laser (Pritel) are filtered via a square-shaped, 5 nm bandwidth filter (Semrock) to eliminate spurious tails at the signal and idler wavelengths, and then injected into the device. A fibre polarisation controller (Lambda) is used to ensure injection of TM0 polarised light to maximise the coupling. After the chip, pump rejection is performed via broadband (>12 nm bandwidth, much larger than the photon spectra) band-pass filters (Opneti), and photons are finally detected using superconducting nanowire single-photon detectors with approximately 80% average efficiency (Photon Spot). For the JSI reconstruction, we use tunable filters with adjustable bandwidth (EXFO XTA-50). Analogue voltage drivers (Qontrol Systems, 300 μV resolution) are used to drive the on-chip phase shifters and reconfigure the integrated circuit.

## Data availability

The data that support the plots within this paper and other findings of this study are available at [https://doi.org/10.6084/m9.figshare.11882760].

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

## Acknowledgements

We thank N. Maraviglia, J. Bulmer and F. Graffitti for useful discussions, and L. Kling for technical assistance. We acknowledge support from the Engineering and Physical Sciences Research Council (EPSRC) Hub in Quantum Computing and Simulation (EP/T001062/1), European Commission QUCHIP (H2020-FETPROACT-3-2014: quantum simulation), QUPIC, and the European Research Council (ERC). L.P. and S.S. have been supported by the University of Trento's strategic initiative Q@TN, the European Union's Horizon 2020 research and innovation programme under grant agreement No. 820405. Fellowship support from EPSRC is acknowledged by A.L. (EP/N003470/1).

## Author contributions

S.P., M.B. and S.S. contributed equally. M.B. conceived the idea of the multi-modal source combined with the delayed pump. M.B. and S.S. performed simulations of the sources and designed the integrated components. S.P., M.B. and S.S. designed the experiments. S.P. and A.M. performed the experiments and analysed the data. L.P. and A.L. supervised the project. All authors contributed to the discussion of the results and to the writing of the manuscript.

## Competing interests

S.P., M.B., S.S. and L.P declare UK patent application number 2005827.7.
