## [Peer Review File · Nature Communications]

Reviewers' Comments:

Reviewer #1:

Remarks to the Author:

The authors propose and experimentally demonstrate an integrated Si photon source exhibiting high visibility and indistinguishability. The authors present a novel pumping approach to yield high purity generation exploiting spontaneous four-wave mixing. The key idea is to use both, dispersion engineering and delayed pump in a multimode four-wave mixing scheme. I believe this work is of great interest for the community. Still, I have some questions regarding the loss inside the chip and the calculation of the heralding efficiency. I consider this work is worth being published in Nature Communications, provided the following comments are properly addressed:

Comment 1:

As stated by the authors on-chip linear loss is key parameter limiting the heralding efficiency. Then, I think it would be necessary to include a detailed description of power loss in each of the building blocks composing the circuit.

Comment 2:

According to the authors, the total insertion loss of the complete chip described in Fig. 2b is approximately 14 dB. Considering an insertion loss of 6.5 dB in the fiber-chip grating couplers, we have a loss of 1 dB within the photonic chip. This 1 dB loss is the total loss experienced by the light after passing through 2 tunable MZI, a 50:50 directional coupler, 2 mode converters and 11 mm of waveguide. Is this correct? If so, these are very impressive results and would recommend the authors to include some further comments in the revised manuscript on the optimization of each building block.

Comment 3:

From table S1 in the supplementary information it seems like the insertion loss of the mode splitter-converter is larger than 12 dB (TE0-to-TE0 -12.67dB, TE1-to-TE1 -12.64 dB). This insertion loss value seems to be inconsistent with insertion loss measurements of the full chip. It would be good to clarify this point. Once the signal and idler photons are generated, they pass through a mode splitter-converter to be split to the two different output ports. Then, having an insertion loss of 12 dB should result in a strong reduction of the heralding efficiency. I think it is very important to well address these points in the revised manuscript.

Comment 4:

I do not understand very well the scheme in Fig. 1H. What is the termination of the bottom output fiber (collecting blue photons)? Is this an absorbing or reflecting termination? If it is a reflecting termination, then do blue photons pass two times through the chip?

Comment 5:

It would be good if the authors included the value of the heralding efficiency when two photon sources are combined in the revised manuscript.

Comment 6:

It would be interesting if the authors included some comments in the revised manuscript on the tradeoffs of using a pulsed source to exploit four-wave mixing in a waveguide versus using a continuous-wave source to exploit four-wave mixing in a resonator. Specially it would be good if the authors commented on the possibility to integrate the pump source together with the Si chip.

Reviewer #2:

Remarks to the Author:

The manuscript submitted by Paesani and colleagues is an impressive piece of work on the

realization of a single photon source in Silicon photonics. They observe a world record purity of a single photon source of 0.9904 for a unfiltered PDC source. They also demonstrate experimentally HOM visibility of 0.96 between two different sources with no photon filtering, that is a remarkable result in its own category.

To my knowledge, no other work shows such high level of performance from unfiltered sources, with other integrated source being either based on resonator with maximum achievable purity of ~93%, or spectrally filtered. Single-emitter/quantum-dots sources have been shown to reach higher purity, but are limited in their indistinguishability, and hence are limited to temporal multiplexing from a single dot.

It is understood that heralded SPDC is the most practical way to generate quantum resources in a photonic circuits, and the properties of the photons are mostly determined by the geometry of the nonlinear interaction. Specifically one would like to generate spectrally uncorrelated photon pairs, that would in turn generate pure heralded single photons after the detection of one of the photons. By properly filtering the interacting fields and tailoring the spectrum both before and during the interaction, one can generate the desired correlation between the photons. There are indeed trade-offs, usually between spectral purity, efficiency and heralding efficiency, but also fabrication achievability and process tolerance. The proposal of Paesani and colleagues is indeed an optimal point and the experimental realization.

I found the work very well written and full of the insights, both in the main matter and in the supplementary materials. There is a copious amount of details regarding the characterization of such high purity, with a very convincing set of multiple experiments. I believe that this submission work both as a record achieving reference as well for the technical discussion of the experiments.

The paper well represents the current ambitions of quantum computation and the paths to overcome the limitations of optical implementations. This work pushes the limits of what is considered achievable on a photonics platform and will be of extreme interest for the many groups that are currently investigating integrated quantum optics.

It may be possible to discuss over the scalability of the approach presented in this manuscript (the main issue I would rise is the reliance on the spectral and temporal preparation of the pump, that could problematic when delivered to a large scale system with many sources and many pump lasers.) but at the end, I believe the most important concern, that is the tolerance of both purity and indistinguishability over process variations, is well addressed. A convincing analysis of the sources of error and possible solutions is given in both the text and the supplemental.

In the end, I strongly recommend the paper for publications, I believe it will set a major milestone in the current high-profile competition between the different approach to the quantum computer.

Reviewer #3:

Remarks to the Author:

The paper presents integrated photon-pair sources on a silicon chip. The key requirements for photon-pair sources in quantum information applications are high purity, high indistinguishability, and high heralding efficiency. With carefully designed multi-mode low-loss waveguides and the dual-spatial-mode pumping technique, the authors demonstrated the generation of two-photon states and heralded single-photon states with remarkably high spectral indistinguishability without using narrowband filtering or resonator configurations (although the source still needs pump rejection filters). Thanks to the low-loss silicon waveguides, on-chip heralding efficiency has also been largely improved compared to other integrated-optics sources.

However, the system heralding efficiency is still insufficient for implementing quantum information

experiments with a large number of single photons due to the large loss in the grating couplers and the pump rejection filters. Moreover, SFWM sources do not produce photons deterministically unlike atomic and quantum-dot single-photon sources. Indeed, it took ~ 4 hours to observe only ~ 100 four-fold coincidence counts in the HOM experiment. Even if the source could be incorporated with perfect detectors and grating couplers as efficient as the one reported in Ref. [32], the system efficiency is still less than 70%, corresponding that the 4-fold coincidence rate is increased to several counts per second. I was wondering what kind of interesting multi-photon experiments can be implemented with this efficiency. Also, the loss of the pump rejection filters is unavoidable in this SFWM source. Although the filters are not on the chip, it is more reasonable to include the filter loss to the loss in the source, not to the characterization loss.

The key techniques for the high single-photon spectral indistinguishability are the dual-spatial-mode pumping and the delay between two nondegenerate pump pulses. However, the techniques have been independently reported previously: a recent demonstration of the dual-spatial-mode pumping scheme should be cited (Feng, L. et al. On-chip transverse-mode entangled photon pair source. NPJ Quantum Inf. 5, 2 (2019)).

In conclusion, despite the quality of the experiment and the manuscript, I am not recommending publication of the paper in Nature Communications, a high-impact multi-disciplinary journal, since some of the key concepts in the source are not new and the improvement in performance is not such that it allows suddenly to realize new kind of experiments with a large number of single photons. The demonstrated improvement of the integrated photon-pair source would attract readers in optics/photonics journals.

Reviewer #1 (Remarks to the Author):

The authors propose and experimentally demonstrate an integrated Si photon source exhibiting high visibility and indistinguishability. The authors present a novel pumping approach to yield high purity generation exploiting spontaneous four-wave mixing. The key idea is to use both, dispersion engineering and delayed pump in a multimode four-wave mixing scheme. I believe this work is of great interest for the community. Still, I have some questions regarding the loss inside the chip and the calculation of the heralding efficiency. I consider this work is worth being published in Nature Communications, provided the following comments are properly addressed:

We thank the reviewer for their time and expertise in carefully reviewing our manuscript, and for their positive view of our work. We are grateful for providing us with valuable and constructive comments, which we have addressed below and in the revised manuscript.

Comment 1:

As stated by the authors on-chip linear loss is key parameter limiting the heralding efficiency. Then, I think it would be necessary to include a detailed description of power loss in each of the building blocks composing the circuit.

We have now expanded the Methods and Supplementary Information 2 to include more details regarding losses for each component employed in our circuit. The characterisation of transmission losses in the multimode waveguides and the grating-coupler efficiencies were already reported in Supplementary Information 2a, as well as the efficiency of off-chip components. We have added Supplementary Information 2c to describe in detail the design and characterisation of the waveguide crossings, which present a loss of approximately 0.4 dB/cross, characterised using the cut-back method. We have added details for mode-converters and directional couplers in Supplementary Information 2b. Simulation results estimate an insertion loss around 0.1 dB and 0.01 dB for the mode converters and the directional couplers respectively. Because this value is significantly smaller than the measurement uncertainty in the test structures (mostly due to the grating couplers), the experimental characterisation provided results with error bars too large to provide a good estimate for such low losses. Nevertheless, this indicates that the low-loss performance prediction from the simulations is reliable, and we thus use simulated results as loss estimates for these components. This discussion and simulation results are now added to Supplementary Information 2b. The estimated losses for all components are now also reported in the Methods of the revised manuscript.

Comment 2:

According to the authors, the total insertion loss of the complete chip described in Fig. 2b is approximately 14 dB. Considering an insertion loss of 6.5 dB in the fiber-chip grating couplers, we have a loss of 1 dB within the photonic chip. This 1 dB loss is the total loss experienced by the light after passing through 2 tunable MZI, a 50:50 directional coupler, 2 mode converters and 11 mm of waveguide. Is this correct? If so, these are very impressive results and would recommend the authors to include some further comments in the revised manuscript on the optimization of each building block.

We estimate approximately 0.5 dB loss in the 11mm multimode waveguide, 0.01dB per directional coupler (therefore 0.02 dB on each MZI composed of two directional couplers), and ~0.1 dB per mode converter, in addition to 0.4 dB in the waveguide crossings. Summing these values, we obtain approximately the 1 dB on-chip transmission loss correctly highlighted by the reviewer.

Such high-performance components are the result of months of design optimisation, and we are pleased the reviewer acknowledges its importance. For example, the directional couplers used on the pump modes, signal photons modes, and idler photons modes, all have slightly different designs to optimise their balanced splitting at the wavelength of the light passing through them (near 1550nm for the pump, near 1588 nm for the signal modes, near 1516 nm for the idler modes). We have now added a discussion and more details in the Supplementary Information 2 regarding the components optimisation performed.

Comment 3:

From table S1 in the supplementary information it seems like the insertion loss of the mode splitter-converter is larger than 12 dB (TE0-to-TE0 -12.67dB, TE1-to-TE1 -12.64 dB). This insertion loss value seems to be inconsistent with insertion loss measurements of the full chip. It would be good to clarify this point. Once the signal and idler photons are generated, they pass through a mode splitter-converter to be split to the two different output ports. Then, having an insertion loss of 12 dB should result in a strong reduction of the heralding efficiency. I think it is very important to well address these points in the revised manuscript.

The insertion losses reported in Table S1 represent the total losses in the mode converter test structure, which include the losses of input and output chip-fiber coupling via grating couplers. As described in the response to Comment 1, losses from the grating couplers dominate the total insertion loss, rendering it difficult to single out the losses from an individual mode-converter in the experiment. In fact, from simulations we estimate losses in the mode converter to be ~ 0.1 dB, which is significantly smaller than the statistical uncertainty on the grating coupler efficiency estimate (± 0.5 dB).

Therefore, while Table S1 conveys two important features for the mode converters, namely the balanced transmission and low cross-talk between the TM0 and TM1 modes, it does not provide an experimental estimation of the intrinsic insertion loss of a single mode-converter. We now report the estimations for insertion losses for the individual mode-converter structure in the Supplementary Information 2.

Comment 4:

I do not understand very well the scheme in Fig. 1H. What is the termination of the bottom output fiber (collecting blue photons)? Is this an absorbing or reflecting termination? If it is a reflecting termination, then do blue photons pass two times through the chip?

In the unheralded correlation measurements (Fig.1H), the termination for the idler (blue) photons is fully absorbing, so that no idler photons are measured nor reflected back into the chip. This is now specified in Fig.1H and in the caption.

Comment 5:

It would be good if the authors included the value of the heralding efficiency when two photon sources are combined in the revised manuscript.

In our experiments we observed no significant change in the heralding efficiency when operating two combined photon sources compared to the single-source operation. In fact, due to the low losses of the additional components used when operating two sources, any difference in the heralding efficiency is expected to be smaller than the uncertainty of the heralding efficiency estimation and thus difficult to observe in our set-up. We now report in the Methods section that the intrinsic heralding efficiencies were observed to be approximately the same in both cases.

Comment 6:

It would be interesting if the authors included some comments in the revised manuscript on the tradeoffs of using a pulsed source to exploit four-wave mixing in a waveguide versus using a continuous-wave source to exploit four-wave mixing in a resonator. Specially it would be good if the authors commented on the possibility to integrate the pump source together with the Si chip.

Continuous wave (CW) excitation imposes a strict frequency anti-correlation between the signal and the idler photons. Hence, irrespective of the device type (resonator or waveguide), this pumping configuration is ineffective for generating uncorrelated pairs as we aim in our work. Moreover, CW operation is not in general viable to perform multi-photon experiments with multiple pairs due to the randomness in the photon emission times: because in the CW pumping regime sources emit photon pairs at random times, the

probability that two or more pairs are generated at the same time is typically negligible. On the other hand, compared to SFWM in waveguides, in the CW regime ring resonators offer superior brightness and improved device miniaturisation [Azzini et al., Optics express 20, 23100 (2012)], which can make them good candidates for two-photon applications where high-rates are required and spectral separability is not needed, such as entanglement-based quantum key distribution protocols (see, e.g., applications in Ref.[9]). In general, as discussed in the Supplementary Information 3a, ring resonators come with trade-offs between brightness and heralding efficiency, and require pump or coupling engineering to emit photons in nearly pure states.

Important steps have been made in the last few years in the field of integrated mode-locked lasers. A promising approach for the integration of pump lasers into our chip is the hetero-integration of electrically pumped III-V materials on a SOI wafer which hosts passive, low-loss components. For example, Davenport et al. [Photonics Research 6, 468 (2018)] have recently reported the hetero-integration of a mode locked laser with 20 GHz repetition rate, 98 mW of peak power and a sech^2 shaped pulse width of 900 fs, which are all values very close to the ones used in our experiment (except the repetition rate, which is much better in the integrated laser).

Discussions on these points are now added in the Supplementary Information 3.

Reviewer #2 (Remarks to the Author):

The manuscript submitted by Paesani and colleagues is an impressive piece of work on the realization of a single photon source in Silicon photonics. They observe a world record purity of a single photon source of 0.9904 for a unfiltered PDC source. They also demonstrate experimentally HOM visibility of 0.96 between two different sources with no photon filtering, that is a remarkable result in its own category.

To my knowledge, no other work shows such high level of performance from unfiltered sources, with other integrated source being either based on resonator with maximum achievable purity of ~93%, or spectrally filtered. Single-emitter/quantum-dots sources have been shown to reach higher purity, but are limited in their indistinguishability, and hence are limited to temporal multiplexing from a single dot.

It is understood that heralded SPDC is the most practical way to generate quantum resources in a photonic circuits, and the properties of the photons are mostly determined by the geometry of the nonlinear interaction. Specifically one would like to generate spectrally uncorrelated photon pairs, that would in turn generate pure heralded single photons after the detection of one of the photons. By properly filtering the interacting fields and tailoring the spectrum both before and during the interaction, one can generate the desired correlation between the photons. There are indeed trade-offs, usually between spectral purity, efficiency and heralding efficiency, but also fabrication achievability and process tolerance. The proposal of Paesani and colleagues is indeed an optimal point and the experimental realization.

I found the work very well written and full of the insights, both in the main matter and in the supplementary materials. There is a copious amount of details regarding the characterization of such high purity, with a very convincing set of multiple experiments. I believe that this submission work both as a record achieving reference as well for the technical discussion of the experiments.

The paper well represents the current ambitions of quantum computation and the paths to overcome the limitations of optical implementations. This work pushes the limits of what is considered achievable on a photonics platform and will be of extreme interest for the many groups that are currently investigating integrated quantum optics.

It may be possible to discuss over the scalability of the approach presented in this manuscript (the main issue I would rise is the reliance on the spectral and temporal preparation of the pump, that could problematic

when delivered to a large scale system with many sources and many pump lasers.) but at the end, I believe the most important concern, that is the tolerance of both purity and indistinguishability over process variations, is well addressed. A convincing analysis of the sources of error and possible solutions is given in both the text and the supplemental.

We are grateful to the reviewer for their time and effort in reviewing our manuscript and for their positive view of our work.

We thank the referee for the interesting comment on how to scale up the pump laser delivery to many sources. In order to stabilize the spectral and temporal preparation of the pump, we envisage an optical circuit in which the same laser is coherently split on-chip by a cascade of MMI devices to synchronously and coherently feed many sources together (see e.g. Ref.[9]). Already at this stage, we were able to deliver a maximum of 15 dBm of average pump power before incurring into detrimental nonlinear pulse distortion from the fiber amplifier. By implementing ultra-low loss grating couplers, as the ones reported in Ref.[33] ([32] of the original manuscript), it would be possible to couple almost 14.5 dBm on chip. This means that, with a single laser, we could simultaneously operate more than 250 sources in parallel with the same level of squeezing as the one implemented in our heralded HOM experiment (see Fig.1E of the main text). As described in the newly added Supplementary Information 4, this number of sources can enable experiments with tens of photons, which are expected to enter computationally interesting regimes with NISQ photonic devices.

In order to further scale to a larger number of sources, a possible solution could be to interconnect the sources with on-chip lasers based on the hetero-integration of electrically pumped III-V materials on a SOI wafer. This possibility is now discussed in Supplementary Information 3b of the revised manuscript.

In the end, I strongly recommend the paper for publications, I believe it will set a major milestone in the current high-profile competition between the different approach to the quantum computer.

Reviewer #3 (Remarks to the Author):

The paper presents integrated photon-pair sources on a silicon chip. The key requirements for photon-pair sources in quantum information applications are high purity, high indistinguishability, and high heralding efficiency. With carefully designed multi-mode low-loss waveguides and the dual-spatial-mode pumping technique, the authors demonstrated the generation of two-photon states and heralded single-photon states with remarkably high spectral indistinguishability without using narrowband filtering or resonator configurations (although the source still needs pump rejection filters). Thanks to the low-loss silicon waveguides, on-chip heralding efficiency has also been largely improved compared to other integrated-optics sources.

We thank the reviewer for their time and effort in carefully reviewing our paper and for providing us valuable and constructive comments which have allowed us to improve the manuscript.

However, the system heralding efficiency is still insufficient for implementing quantum information experiments with a large number of single photons due to the large loss in the grating couplers and the pump rejection filters. Moreover, SFWM sources do not produce photons deterministically unlike atomic and quantum-dot single-photon sources. Indeed, it took ~4 hours to observe only ~100 four-fold coincidence counts in the HOM experiment. Even if the source could be incorporated with perfect detectors and grating couplers as efficient as the one reported in Ref. [32], the system efficiency is still less than 70%, corresponding that the 4-fold coincidence rate is increased to several counts per second. I was wondering what kind of interesting multi-photon experiments can be implemented with this efficiency.

We are grateful to the reviewer for raising an important point on the experiments that are enabled by our source in the near-term. We have added a new supplementary section (Supplementary Information 4 in the updated manuscript) to investigate and discuss what performance we can expect using our sources in near-term noisy intermediate scale quantum (NISQ) photonic devices, focusing on photonic sampling. The results in terms of rates for multi-photon experiments are reported in the newly added Table S2 and Fig. S9. In the latter we also compare our sources with the estimations from previous state-of-the-art integrated photon sources. To reflect the capability of currently available integrated technologies, for all the relevant parameters in these simulations we use exactly the values characterised in our experimental set-up, with the only exception of the grating couplers, for which we use the efficiencies as in Ref.[33] ([32] of the original manuscript, as suggested in the reviewer's comment). It can be observed that, using current components and off-chip detectors, our sources enable experiments with ≥ 20 photons using arrays of ≥ 32 sources, well within the current fabrication capabilities of silicon photonics, which would otherwise be impossible with previous integrated photon sources (for which the expected rates are several orders of magnitude lower). For example, in the 32 sources case we estimate 4-photon events at a MHz rate, while 20-photon events with a rate of approximately one event per hour. Implementing approximately 125 sources, 20-photon events are expected at Hz rates, and experiments towards 30 photons become available. Such experiments are expected to enter into the regime of quantum advantage, where quantum machines can surpass classical supercomputers (see e.g. Ref.[39]), and would represent a major milestone for photonic quantum computing.

Moreover, the quality of the generated photons is also an important factor for NISQ photonic devices. Recent works (see Ref.[17]) have highlighted that imperfections of the photons, e.g. due to non-unit purity and distinguishability, imply a photon number threshold above which quantum photonic sampling experiments can be efficiently simulated on classical computers. That is, imperfect sources pose a limit to the computational complexity of photonic experiments independently on how many photons they are able to generate. We expanded this discussion, which is also mentioned in the conclusion of the original and updated manuscript, in the newly added Supplementary Information 4. The thresholds estimated for our sources and, for comparison, for state-of-the-art on-chip sources and quantum dots emitters, are now reported in Fig. S10. The simulated results show that in this metric our sources are compatible with genuinely quantum experiments with several tens of photons, significantly improving previous capabilities.

In conclusion, leveraging current silicon photonic technologies, our sources are expected to enable near-term integrated quantum photonics experiments with ≥ 20 high-quality photons which genuinely do not allow efficient classical simulations. Multi-photon experiments at this scale are expected to enter computational interesting regimes where quantum machines can compete and surpass classical supercomputers [Neville et al., Nature Physics 13, 1153 (2017)], with possible industrially relevant applications (e.g. quantum chemistry simulations [Huh et al., Nature Photon 9, 615 (2015); Sparrow et al., Nature 557, 660 (2018)]).

In general, by simultaneously satisfying all properties required for sources compatible with scalable architectures for photonic quantum computing (i.e. mass-manufacturability, high purity, high indistinguishability, and high heralding efficiency) our sources represent a significant step towards a technology that, in the long term, may ultimately lead to a universal photonic quantum computer. We believe this prospect is on its own a valid ground to support the impact of our results.

Also, the loss of the pump rejection filters is unavoidable in this SFWM source. Although the filters are not on the chip, it is more reasonable to include the filter loss to the loss in the source, not to the characterization loss.

To the best of our knowledge, the standard in the literature on spontaneous photon sources is to not include the transmission loss of the pump rejection filters (as long as they do not affect the photon spectral properties, as in our case) in the intrinsic heralding efficiency of the source. A first reason why we do not include them when calculating the heralding efficiency is thus consistency with the literature and to allow a fair comparison with previously reported spontaneous photon sources.

Furthermore, we believe it should be more accurate to account pump-rejection filters as part of the detection apparatus rather than the source itself: their use is to perform spectrally-selective detection at the wavelength range of the photons, excluding the range associated to the pump. While this spectral selection is required to remove noise when performing single photon detection, we remark it is not necessarily the case for any detection apparatus. For example, if homodyne detection (Gaussian measurement) is used to measure the output modes of our device rather than threshold photon detectors (non-Gaussian measurement), then the spectral selection can be performed by the choice of the local oscillator wavelength, with no (or much more limited) use of pump rejection filters. Therefore, strictly speaking, we believe the use of pump rejection filters is not an intrinsic requirement of SFWM sources, and that it is thus legitimate to not include it in the intrinsic heralding efficiency of a spontaneous photon source.

In any case, we understand the reviewer's point, and agree that in most practical applications of our sources pump rejection filters will be required, and will affect the total heralding efficiency. For completeness the heralding efficiency value corrected with the insertion loss for the pump rejection filters used (which is already quite moderate ~ 0.4 dB, and could be readily improved to < 0.1 dB by removing fiber-to-fiber interconnection through fiber splicing) is now also reported in the Methods, and corresponds to 83%.

The key techniques for the high single-photon spectral indistinguishability are the dual-spatial-mode pumping and the delay between two nondegenerate pump pulses. However, the techniques have been independently reported previously: a recent demonstration of the dual-spatial-mode pumping scheme should be cited (Feng, L. et al. On-chip transverse-mode entangled photon pair source. NPJ Quantum Inf. 5, 2 (2019)).

The key technique that enables our near-ideal photon sources is the creative combination of both the phase-matching engineering in intermodal SFWM and the delayed-pump strategy. This is a fundamental point because, despite these approaches having been individually investigated in optical fibres or integrated devices in previous works, it is only their innovative combination that enables on-chip high-quality photon sources. To the best of our knowledge, ours is the first work that explores this new idea, and demonstrates it achieving near-ideal integrated photon sources in Silicon quantum photonics.

The work of Feng et al. mentioned by the reviewer, now cited in the revised manuscript, reports interesting results for the on-chip generation of transverse-mode photonic entanglement. While the design in Feng et al. may look similar to ours (i.e. both designs are based on mode-converters and multi-mode waveguides), the techniques used are quite different. We highlight two important differences which, amongst others, we think are particularly significant. 1) Techniques to generate high-purity heralded photons are not investigated by Feng et al. In particular, while intermodal SFWM is used for the generation of entanglement, no phase-matching engineering is performed on their device. In our work, a sophisticated tailoring of the intermodal phase-matching to produce discrete phase-matching bands far from the pump was key to achieve uncorrelated photons. This or similar techniques are not presented in Feng et al., and strong spectral correlations should instead be expected from their sources. In fact, Feng et al. do not report any evidence that their device could improve the single photon spectral purity without the use of filters, which is instead the main outcome of our study. 2) The delayed-pump technique is completely absent in Feng et al..

In conclusion, the technique we develop, where tailored phase-matching conditions are combined with the delayed-pump strategy to generate spectrally uncorrelated photons, is very different from what has been reported before in integrated photonics, and represents an important novelty of this work. No previously reported technique has opened the possibility to simultaneously satisfy all the requirements for on-chip photon sources compatible with scalable quantum photonic architectures: the novel ideas developed and implemented in this work have been crucial to achieve this milestone.

In conclusion, despite the quality of the experiment and the manuscript, I am not recommending publication of the paper in Nature Communications, a high-impact multi-disciplinary journal, since some of the key

concepts in the source are not new and the improvement in performance is not such that it allows suddenly to realize new kind of experiments with a large number of single photons. The demonstrated improvement of the integrated photon-pair source would attract readers in optics/photonics journals.

We again thank the reviewer for their time and effort in reviewing our paper. We believe their concerns have been addressed here and in the revised manuscript. We hope the novelty and impact of our results is now clearer.

Reviewers' Comments:

Reviewer #1:

Remarks to the Author: I am fully satisfied by the responses provided by the authors. I highly recommend this paper for publication in Nature Communications.

Reviewer #3:

Remarks to the author: I appreciate the authors very much for addressing my comments; now the authors' comments and revisions make it clear how important the techniques for producing indistinguishable photons with the integrated source is.

The new supplementary section 4 discussing the near-term scalability with the presented integrated source is also nice, but I have several questions/comments about the section: I tried to reproduce the numbers in Table S2 using Eq. S26 and parameters shown in the section ($R_0 = 5 \times 10^7$, $|\tanh(\xi)|^2 = 0.03$, $\eta_{gc} = 0.89$, $\eta_{ch} = 0.8$, $\eta_{det} = 0.8$, $\eta_u = 0.995$, $\eta_{her} = 0.9$), but my calculation is not matched to the numbers in the table. For example, for 4 sources ($k = 4$) and 2 heralded photons ($n = 4$),

$$\begin{aligned} R(n = 4, k = 4, m = k = 4) &= R_0(k, n/2) \tanh(\xi)^n \operatorname{sech}(\xi)^k \eta_u^{mn/2} (\eta_{her} \eta_{gc} \eta_{ch} \eta_{det})^n \\ &= 5 \times 10^7 \times 6 \times 0.03^2 \times (1-0.03)^2 \times 0.995^{4 \times 4/2} \times (0.9 \times 0.89 \times 0.8 \times 0.8)^4 \\ &= 1.7 \times 10^4. \end{aligned}$$

This is different from the one shown in Table S2 (5×10^4). I also tried several different parameters but none of them reproduced all of the numbers in Table S2 simultaneously. I recommend the authors to recheck the calculations. Also, the exponent of $\operatorname{sech}(\xi)$ in Eq. S26 would be $2k$, not k (although this minor change does not explain the difference).

Related to the above comment, I also want to confirm if the authors used $|\tanh(\xi)|^2 \sim 0.03$ in the experiments. I doubt this because the heralded $g_h^{(2)} = 0.0056$ shown in the main text is too low compared to the squeezing parameter. For $|\tanh(\xi)|^2 \sim 0.03$, the single- (double-)pair generation probability $P_1(P_2)$ is approximately $0.03(0.03^2)$. Therefore, the $g^{(2)} = \langle a^\dagger{}^2 a^2 \rangle / \langle a^\dagger a \rangle^2 \sim 2P_2/P_1^2 = 0.06$, for the perfect single-photon (non-number-resolving) detectors (and this becomes higher for inefficient single-photon detectors). The authors might have used much lower

pump power (~ 0.05 mW) compared to the one shown in the main text (0.5 mW) for the HOM experiment. If this is true, Fig. 1E and Table S2 should be corrected and the text should be modified accordingly.

Now I think that the presented work is worth for publication, but suggest the authors to address the above comments and to make corrections, if necessary.

Reviewer #3 (Remarks to the Author):

I appreciate the authors very much for addressing my comments; now the authors' comments and revisions make it clear how important the techniques for producing indistinguishable photons with the integrated source is.

The new supplementary section 4 discussing the near-term scalability with the presented integrated source is also nice, but I have several questions/comments about the section: I tried to reproduce the numbers in Table S2 using Eq. S26 and parameters shown in the section ($R_0 = 5 \times 10^7$, $|\tanh(\xi)|^2 = 0.03$, $\eta_{gc} = 0.89$, $\eta_{ch} = 0.8$, $\eta_{det} = 0.8$, $\eta_u = 0.995$, $\eta_{her} = 0.9$), but my calculation is not matched to the numbers in the table. For example, for 4 sources ($k = 4$) and 2 heralded photons ($n = 4$),

$$\begin{aligned} R(n = 4, k = 4, m = k = 4) &= R_0(k, n/2) \tanh(\xi)^n \operatorname{sech}(\xi)^k \eta_u^{m/2} (\eta_{her} \eta_{gc} \eta_{ch} \eta_{det})^n \\ &= 5 \times 10^7 \times 6 \times 0.03^2 \times (1-0.03)^2 \times 0.995^{4 \times 4/2} \times (0.9 \times 0.89 \times 0.8 \times 0.8)^4 \\ &= 1.7 \times 10^4. \end{aligned}$$

This is different from the one shown in Table S2 (5×10^4). I also tried several different parameters but none of them reproduced all of the numbers in Table S2 simultaneously. I recommend the authors to recheck the calculations. Also, the exponent of $\operatorname{sech}(\xi)$ in Eq. S26 would be $2k$, not k (although this minor change does not explain the difference).

We are very grateful to the referee for their careful review, and for spotting an erratum in this supplementary section prepared in the first revision. The simulation results that were reported used a squeezing value $|\tanh(\xi)|^2 = 0.05$, while the value mentioned in the text was 0.03, which was causing the confusion correctly pointed out by the reviewer. We have now updated this section with simulation results using $|\tanh(\xi)|^2 = 0.03$ as in the text. For the $k = 4$, $n = 4$ case we obtain a rate of 1.6×10^4 Hz, consistent with the reviewer's calculation. As can be observed in the updated Supplementary Table 2, also with this squeezing value 20-fold events are accessible at rates of approximately 0.5 event/minute when $k=128$. The main conclusion of this analysis, namely that on-chip experiments with ≥ 20 photons are enabled with our sources in the current silicon photonics technology, remains correct. In the updated version we also corrected the second typo pointed out by the reviewer: the exponent of $\operatorname{sech}(\xi)$ in Supplementary Eq. 30 (Eq. S26 in the previous version) has now been corrected to $2k$ instead of k .

Related to the above comment, I also want to confirm if the authors used $|\tanh(x)|^2 \sim 0.03$ in the experiments. I doubt this because the heralded $g_h^{(2)} = 0.0056$ shown in the main text is too low compared to the squeezing parameter. For $|\tanh(\xi)|^2 \sim 0.03$, the single- (double-)pair generation probability $P_1(P_2)$ is approximately 0.03(0.032). Therefore, the $g(2)$ conditioned by idler photon detection is $g^{(2)} = \langle a^{\dagger 2} a^2 \rangle / \langle a^{\dagger} a \rangle^2 \sim 2P_2(P_1 + P_2) / P_1^2 = 0.06$, for the perfect single-photon (non-number-resolving) detectors (and this becomes higher for inefficient single-photon detectors). The authors might have used much lower pump power (~ 0.05 mW) compared to the one shown in the main text (0.5 mW) for the HOM experiment. If this is true, Fig. 1E and Table S2 should be corrected and the text should be modified accordingly.

We thank the reviewer again for his careful examination. We indeed found that an incorrect offset was present in the x axis of Fig.1E inset, which has now been corrected and the text updated with the correct value of the g_2 (0.053).

Now I think that the presented work is worth for publication, but suggest the authors to address the above comments and to make corrections, if necessary.

We thank the reviewer for their time and effort in carefully reviewing our paper, which allowed us to improve and correct it.